# Effects of Marketing Decisions on Brand Equity and Franchise Performance

**Eunkyung Lee** [1], **Ji-Hern Kim** [2,*] **and Chang Seop Rhee** [2]

1   International Business School Suzhou, Xi'an Jiaotong-Liverpool University, Suzhou 215123, China;
    Eunkyung.Lee@xjtlu.edu.cn
2   Department of Business Administration, School of Business and Economics, Sejong University,
    Seoul 05006, Korea; crhee2@sejong.ac.kr
*   Correspondence: jihern@sejong.ac.kr; Tel.: +82-2-3408-3171

**Abstract:** The purpose of this study is to provide a way of pursuing a balanced profitability between franchisors and franchisees leading to the sustainability in franchising. Based on a belief that the formation of brand-centric relationship is vital for the success of franchising system, we constructed a model that examines the relationships between marketing decisions, brand equity, and the financial performances of franchisors and franchisees. We used actual data of the Korean franchise chains, including measures of channel intensity and advertising and promotional activities as franchise marketing decisions as well as the profitability of franchisors and franchisees for the analysis. The results of analysis show that while advertising and promotion expenditure has a positive impact on the performances of both franchisors and franchisees, the number of stores does not influence them in the same way. This indicates that their interests may conflict. This study suggests that marketing decisions can be utilized as a means of achieving balanced profitability that would benefit the sustainability in franchising between franchisors and franchisees.

**Keywords:** franchise; brand equity; franchise performance; franchise relationship; marketing decisions





## 1. Introduction

In franchising, two firms form a relationship based on a licensing agreement. In this relationship, a franchisee acquires the rights from a franchisor to sell its goods or services with its brand name along with the support of business resources in return for entry fees and royalty payments [1]. Through these agreements, franchisees can take advantage of business experiences and systems of the franchisor while the franchisor can accelerate the business expansion via franchisees without adding the burden of directly opening and running new outlets [2,3].

At the center of the success of a franchise is the relationship between a franchisor and its franchisees. The economic interests of franchisors and franchisees are highly vulnerable to each other due to their contractual relationships [4]. For example, franchisees' performance would suffer if a franchisor does not provide the corporate support it promised in exchange for the royalty fees. Conversely, franchisees can harm a franchisor if they pursue their margins while sabotaging the franchise brand. Thus, understanding the nature of franchise relationships and effectively managing the relationships are keys to a successful and sustainable franchise system [5].

A considerable amount of past literature has focused on investigating factors that drive effective management of franchise relationships [6]. The discussions in this stream of literature mainly pertain to governance issues including decisions about managing franchisees to reduce free-riding [7]. These studies are mainly limited to the franchisors' perspective, focusing on how to influence and control franchisees in the interests of franchisors rather than franchisees. Taking a step further, recent literature suggests that pursuing balanced profitability for both franchisors and franchisees would increase the chances of survival for

the franchise operation and lead to a sustainable franchise relationship by incentivizing both parties to work together [1,8]. Especially, Badrinarayanan et al. (2016) stress that the cultivation of a brand-centric relationship is vital for the success of the franchising system because brand equity is highly related to the profitability for both franchisors and franchisees [9]. Specifically, franchisees enter the franchise network to benefit from the franchise chain's brand equity whereas franchisors expect to gain from the royalty fees from franchisees in exchange for their use of brand equity [10,11]. Accordingly, two parties need to share the responsibility to manage brand equity as an intangible resource and cooperate to gain more profitability [12].

Past research regarding brand equity shows that marketing decisions on channel intensity or advertising and promotional activities can affect corporate performance directly as well as indirectly through brand equity [13–15]. In other words, marketing decisions made by a franchisor may influence the equity of the franchise brand and ultimately affect the profitability of a franchisor and a franchisee. However, the valence and the magnitude of impact that marketing decisions have on the franchisor and franchisee may not be the same—some marketing decisions may influence both parties while some may not, some marketing decisions may influence both parties in the same direction while some may not. Building upon the notion that balanced profitability for two parties is essential for sustaining a healthy franchise relationship, it is necessary to understand how each of the marketing decisions affects brand equity and the performance of franchisors and franchisees [1]. However, as far as we know, there is no research to address this issue.

Given the above discussion, we establish and test a comprehensive model that examines the relationship between marketing decisions, brand equity, and the financial performances of franchisors and franchisees. We conduct an analysis using actual data of the Korean franchise chains including measures of channel intensity and advertising and promotional activities as franchise marketing decisions as well as the profitability of franchisors and franchisees. As the Korean franchise industry is rapidly developing in East Asia, it will be a good research setting to study the sustainability of franchise chains [16,17]. Unlike extant research focusing on the only franchisors' performance (e.g., Wang et al., 2020), this research analyzes a separate set of information as the financial performances of franchisors and franchisees [11]. It would allow us to examine how franchisors and franchisees are influenced differently by channel intensity, or advertising and promotional activities. The findings of this research can provide practitioners in a franchise industry with meaningful insight into marketing decisions for a sustainable franchisor–franchisee relationship.

## 2. Theoretical Background

The central issue in the field of franchising is effectively managing relationships between franchisors and franchisees to develop a sustainable and healthy franchise system [1,18]. Because the interests of franchisors and franchisees are intertwined in a contractual relationship, franchising not only brings benefits to both parties but also increases the risk and vulnerability as well [19]. Furthermore, there is an asymmetry of information and knowledge about the market and the franchise operation between franchisors and franchisees, which can also bring misalignment of interests in a franchise relationship [20]. It can cause the dispute between two parties, which leads to the litigation to have a negative influence on the franchise system [11].

For reducing potential conflict of interests and achieve a win-win result, recent research has emphasized the importance of building a franchisor–franchisee relationship based upon brand equity [9]. In this line of research, it is suggested that the two parties—the franchisor and franchisee—should share the responsibility of cultivating brand equity because it would benefit them both with greater profit. Brand equity refers to the values added by a brand that drives consumers' differential responses to the marketing actions [21,22]. Subsequently, the higher the brand equity, the more effective marketing actions are. Brand equity is a multidimensional concept and, thus, various compositional factors are considered by

past research [23,24]. Especially, since Keller (1993), which proposed brand awareness and brand image as a set of core factors that construct brand equity, a considerable number of past researches has used them to measure brand equity and examined its antecedents and consequences [25–27]. In this stream of research, brand equity is predicted to be greater when consumers are well aware of a brand and have many positive associations.

The framework that is often used to describe the role of brand equity in driving firm performance is the model of Brand Value Chain [28]. The model suggests that various marketing decisions that firms make generate brand equity and ultimately influence market performance, such as profitability or market share, by driving consumer purchases [29,30]. In other words, the model describes a chain of events that starts from firms' marketing decisions to brand equity development and ends with market performance generation. Prior literature has examined the presence of such a process around brand equity not only in the business-to-customer (B2C) but also in the business-to-business (B2B) industries as well [15,31,32].

In the franchise context, the marketing decisions that franchisors make for the franchise chain and its brand would be analogous to the steps of this model. Decisions on how many franchise outlets to operate or how much to spend on advertising and promotional activities are part of the marketing decisions that franchisors make [33,34]. Such decisions would subsequently influence the market performances of franchisors and franchisees by enhancing brand equity and driving consumer purchases. However, what is unique about the franchise relationship is that there are two closely related parties involved in the franchise relationship and marketing decisions may not influence them in the same manner [1,35]. Thus, it is necessary to delineate the impacts of franchise marketing decisions on franchisors and franchisees, which the result would show us whether franchisors' decisions about the franchise operation manifests in balanced profitability for both parties.

Building upon the key components that we have identified, we develop a model that examines the relationships between marketing decisions on channel intensity and advertising and promotional expenditure, brand equity, and financial performances of both franchisees and franchisors. In the following section, we derive hypotheses based on the general belief that is supported by extant research. Then, based on the unique nature of the franchising system, we suggest some counterarguments about the previous reasoning, which show the importance of testing the hypotheses again in a franchise context.

## 3. Hypothesized Model

### 3.1. Relationships between Marketing Decisions and Brand Equity

3.1.1. The Effect of Number of Total Stores on Brand Equity

The decision about the number of chain outlets to operate is an important component of franchise marketing decisions, representing the channel intensity of the franchise operations. In general, an increase in the number of stores for a company or a brand would result in a greater chance of coming into contact with potential customers (i.e., brand awareness) and communicating information about the brand (i.e., brand image) [36,37]. Subsequently, consumers are more likely to perceive greater value for the brand and their perception of brand equity would be greater [15,37]. This general relationship would also be present in the franchise relationship because the manner in which consumers come into contact with the franchise chain is similar.

However, at the same time, the growth in the size of the franchise chain may also challenge franchisors' control over the franchise brand because it becomes more difficult to maintain brand consistency across franchisees that have a degree of autonomy over their operations [38,39]. Considering that franchisees may not always adhere to the franchisors' decisions about the franchise brand, the risk of harming the brand equity of the franchise brand may rise as the number of stores in operation increases for the franchise chain [1,40]. Thus, it is necessary to test whether the general belief in how the number of total stores would influence brand equity in the franchise context as well. Given the above discussion, the following hypothesis is derived.

**Hypothesis 1a.** *The number of total stores has a positive effect on brand equity.*

3.1.2. The Effect of Advertising and Promotion Expenditures on Brand Equity

Advertising and sales promotions are the most commonly used marketing communication tactics for enhancing brand equity [41]. Firms allocate a significant portion of their budgets to advertising and promotions to increase brand exposure and to provide customers with information about the brand. Marketing decisions regarding advertising and promotion expenditures in the franchise context is mostly made by franchisors and franchisees are often pressured to follow the decisions made by the franchisors to communicate consistent messages about the franchise brand [1]. If the franchisees decide to adhere to the franchisors' decisions about advertising and promotional activities, then the brand equity of the franchise brand would increase as consistent brand messages are communicated throughout the franchise chain [15,37].

However, because of the unique nature of the franchise context in which franchisees also have a degree of control over its operation, it is still possible that franchisees may not decide to participate in the campaigns [5,40]. In this case, an increase in spending on advertising and promotional activities would not contribute to enhancing brand equity because the brand cannot be communicated consistently [38]. Hence, it is necessary to investigate whether the general belief about the relationship between advertising and promotion expenditures and brand equity also holds in the franchise context. This leads to the following hypothesis 1b.

**Hypothesis 1b.** *The advertising and promotion expenditure has a positive effect on brand equity.*

*3.2. Relationships between Brand Equity and Performances of Franchisors and Franchisees*

It is the general belief that brand equity has a positive impact on firms' performances [13,42]. Specifically, when a brand is low in brand awareness, it is unlikely for the brand to be included in the consumers' consideration sets. Without entering the consideration sets of potential customers, firms cannot generate sales and revenues. Similarly, when a brand has a positive image that differentiates it from competitors, consumers are more likely to have a higher purchase intention for the brand. Such a relationship would also hold in the franchise context because potential franchisees would seek contracts with franchises with strong brands with the expectation of generating a greater amount of revenue [1,43]. Given the above discussion, the following hypotheses 2a and 2b are derived.

**Hypothesis 2a.** *Brand equity has a positive effect on the average annual revenue of franchisees.*

**Hypothesis 2b.** *Brand equity has a positive effect on the annual revenue of the franchisor.*

*3.3. Relationships between Marketing Decisions and Performances of Franchisors and Franchisees*
3.3.1. The Effect of Number of Total Stores on Franchise Performance

The characteristics of marketing decisions that franchisors conduct can also directly influence the financial performances of franchisors and franchisees. From the franchisors' perspective, past literature has shown that the size of the franchise system, represented by the total number of stores in the franchise network, has a positive effect on the performance of franchisors [43,44]. An increase in the number of total stores means that franchisors can collect more royalty fees from a greater number of franchisees, which is the major source of revenue for franchisors. Furthermore, they can benefit from achieving economies of scale in the operation of franchise networks as the number of stores increases [6]. For example, franchisors that supply raw materials or ingredients to franchisees can save costs in purchasing if they have a large group of franchisees.

Meanwhile, from the franchisees' perspective, consumers would be more likely to purchase products of franchise brands that they encounter most frequently because they consider such frequent contact as a signal of popularity. According to the literature on brand

popularity, consumers are likely to perceive popular brands to have lower risks of failure, as they are already chosen by other consumers [45,46]. As a result, they use a popularity cue as a decisional shortcut because it has an advantage of information processing. Moreover, consumers tend to believe that brand popularity results from high quality because firms have a chance to receive and reflect the feedback from many customers and are forced to improve their goods and services [47].

However, there may also be a greater risk of power shifting towards the franchisees as the size of the franchisee network grows [48]. For example, franchisees can ally to maximize their negotiating power, resulting in contract terms that are less favorable for the franchisors. Moreover, the risk of performance reduction for the franchisees may be greater when the number of stores in the franchise chain increases because the internal competition would rise as well [49]. They would need to compete against other franchisees within the vicinity over consumers' share of pocket. Thus, it is necessary to closely investigate the directionality of effect that the number of stores has on the financial performance of franchisors and franchisees. Given the above discussion, the following hypotheses 3a and 3b are derived.

**Hypothesis 3a.** *The number of total stores has a positive effect on the average annual revenue of franchisees.*

**Hypothesis 3b.** *The number of total stores has a positive effect on the annual revenue of the franchisor.*

3.3.2. The Effect of Advertising and Promotion Expenditures on Franchise Performance

In terms of the advertising and promotional expenditures, it is a general belief that such spending positively influences the firm performance and ultimately, firm values [50–53]. A similar prediction would also hold in the franchise context, since spending on advertising and promotional activities, such as free gifts or price discounts would trigger consumer purchase, especially in the short run [53]. Subsequently, the sales and revenues of franchisees would increase while the franchisor would also benefit from the increase in royalty payment from franchisees since because the amount of royalty fee payment is usually determined as a portion of franchisees' revenue or profit [54].

However, considering the contractual nature of the franchise relationship, it is possible that the increase in advertising and promotional expenditures may not always contribute to the performance growth for franchisors and franchisees. Most of the marketing decisions regarding advertising and promotional expenditures are centrally determined by the franchisors. Because franchisors often allocate a portion of the advertising and promotional expenditures to franchisees, their cost burden is greater [55]. With the burden, it is likely that franchisees may not commit to the advertising and promotional activities, resulting in lower performance. Franchisors' performance may also be negatively influenced by an increase in the advertising and promotion expenditures when the efficiency of promotional activity is low. A decrease in the franchisees' performance would further lead to a negative impact on franchisors' performance as well. Thus, it is necessary to investigate the directionality of the influences that advertising and promotional expenditures have on franchisees' and franchisors' financial performances. This leads to the following hypotheses 4a and 4b.

**Hypothesis 4a.** *The advertising and promotion expenditure has a positive effect on the franchisees' financial performance.*

**Hypothesis 4b.** *The advertising and promotion expenditure has a positive effect on the franchisors' financial performance.*

### 3.4. Relationships between Performances of Franchisors and Franchisees

The nature of the relationship between franchisors and franchisees involves economic dependence between the two parties [4,6]. The contract between franchisors and franchisees specifies how franchisors and franchisees are going to share the profits from the franchise operations. One of the most commonly used profit models adopted by the franchising industry involves royalty fees from franchisees for the right to use their brand names and the entry fees that franchisees must pay to open an outlet for the franchise brand. The amount of royalty fees that franchisees must pay to the franchisor each year is often determined as a percentage of sales or profits that the franchisees earn from their services [1,54]. In other words, if the franchisees run successful operations, then their success would contribute to the franchisor's performance through the increase in the amount of royalty fees that franchisors can collect from franchisees. This leads to the following Hypothesis 5. The hypotheses discussed so far are summarized in Figure 1.

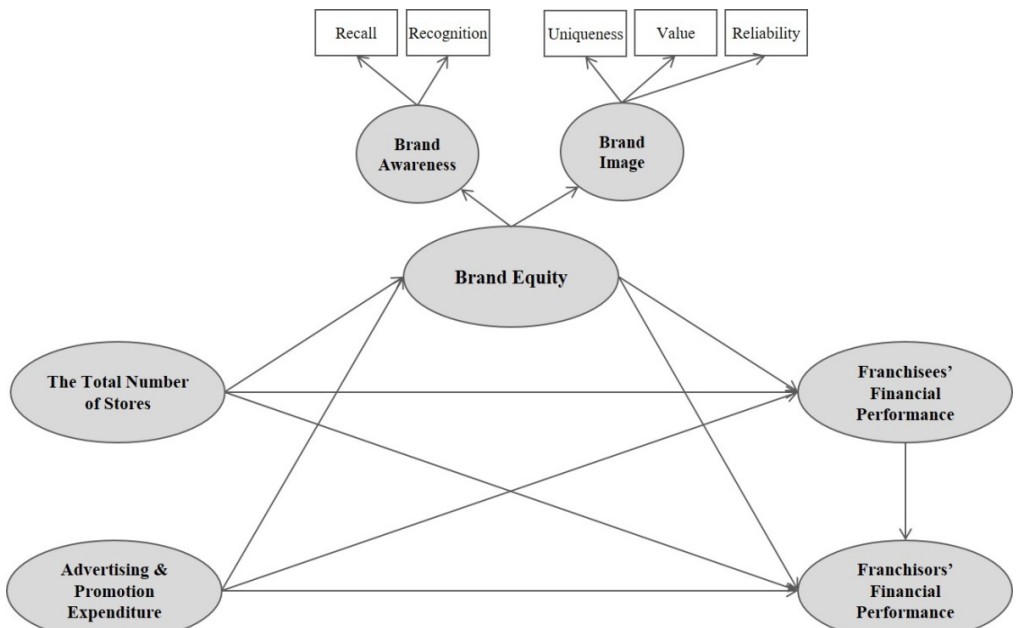

**Figure 1.** Structural relationships among franchise characteristics, brand perception, and financial performances.

**Hypothesis 5.** *The financial performance of franchisees has a positive effect on the financial performance of franchisors.*

### 4. Method

#### 4.1. Sampling and Data Collection

The empirical setting for this research was focused on the franchise industry in Korea. Specifically, we focus on the franchise businesses in the service sectors, ranging from food and beverages to health and beauty care services. To test the hypothesized model, we gathered data from two different sources from 2012 through 2017. First, the information regarding the franchise marketing decisions, including the total number of stores and advertising and promotion expenditures, and the financial performance information for both franchisors and franchisees were taken from the Franchise Disclosure Document announced by the Korean Fair Trade Commission (KFTC). The financial performance data was cross-checked with the annual reports from the KIS-VALUE database.

Second, we collected information on consumers' brand perception from the Korea Brand Power Index (K-BPI) announced by Korea Management Association Consulting (KMAC), which is one of the largest Korean consulting companies that focuses on business

analysis and corporate management training and consulting. Every year, KMAC conducts interviews with 12,000 adults regarding their brand perception and attitude towards brands operating in Korea and creates an index that indicates brand equity. Specifically, KMAC collects brand data on items from Keller (1993) that defines brand equity in the dimensions of brand awareness and brand image [24].

Using the information derived from two sources, we compared and extracted brands that were matched across all three data sets (marketing decision, brand equity, franchise performance) for each year. After eliminating data with missing information on key variables, we were left with a total of 256 data. Table 1 summarizes the basic description of the data samples used for the analysis.

**Table 1.** Basic descriptions of the data sample.

| Industry | 2012 | 2013 | 2014 | 2015 | 2016 | 2017 | Freq. | Per. |
|---|---|---|---|---|---|---|---|---|
| Tire Shop | | | | | 1 | | 1 | 0.39 |
| Supermarket | | | | 4 | 4 | 4 | 12 | 4.69 |
| Convenience Store | 3 | 3 | 2 | 1 | 5 | 4 | 18 | 7.03 |
| Living SPA | | | | | | 1 | 1 | 0.39 |
| Shoe Store | | | | 1 | 1 | 1 | 3 | 1.17 |
| Health and Beauty Store | | | 1 | 1 | 1 | 1 | 4 | 1.56 |
| Cosmetic Brand Shop | 5 | 6 | 4 | 7 | 6 | 7 | 35 | 13.67 |
| Fashion Jewelry Shop | | 1 | | 1 | 2 | 1 | 5 | 1.95 |
| Korean Food Restaurant | | | | 1 | 1 | 1 | 3 | 1.17 |
| Donut Shop | 1 | | | 1 | | | 2 | 0.78 |
| Bakery | 1 | 3 | | 2 | 3 | 3 | 12 | 4.69 |
| Fast Food Chain | 1 | 2 | 1 | 2 | 3 | 3 | 12 | 4.69 |
| Pizza Chain | 4 | 3 | 2 | 3 | 10 | 8 | 30 | 11.72 |
| Chicken Chain | | 5 | 8 | 10 | 13 | 12 | 48 | 18.75 |
| Ice Cream Chain | 2 | | | 1 | 4 | | 7 | 2.73 |
| Coffee Shop | 3 | 3 | | 6 | 7 | 7 | 26 | 10.16 |
| Educational Service | | 2 | | 3 | 8 | 9 | 22 | 8.59 |
| Auto Repair Shop | | | | 2 | 1 | 1 | 4 | 1.56 |
| Hair Shop | | | | 3 | 4 | 4 | 11 | 4.30 |

*4.2. Variables*

4.2.1. Franchise Marketing Decisions

We collected the data on the number of total stores and advertising and promotion expenditure from the Franchise Disclosure Document to examine their effects on consumers' brand perception and revenue. The number of total stores was operationalized as the total number of stores each year. We operationalized the marketing and promotion expenditures as the total amount of annual spending for both a franchisor and franchisees on marketing and promotion activities, which were specified by the disclosure document. We further confirmed the data with the marketing and promotion expenditure information included in annual reports and audit reports from KIS-VALUE. Both variables were log-transformed because the variance of numbers significantly varied across industries and firms.

4.2.2. Brand Equity

From the items included in K-BPI, we extracted items on brand awareness and brand image to measure brand equity [24]. Brand awareness and brand image are not only the key dimensions that construct brand equity but also the key components included in the Brand Value Chain to capture the brand equity. Specifically, we selected two items, brand recall and recognition, to measure brand awareness and three items of brand image, perceived value over price, uniqueness, and reliability, for analysis based on this definition. Brand recall was measured as the percentage of respondents who mentioned a particular franchise brand without any cues. Brand recognition was measured as the percentage of respondents who said they recognized the franchise brand when they were presented with the brand name. The brand image was measured with three items using a seven-point

Likert scale (To what extent do you agree with the following statement, "Brand A has a good value considering its price", "Brand A is unique", and "Brand A is reliable"). In particular, when we examined the skewness of the two items measuring brand awareness, both items exhibited a strong negative skew. This implied that most of the brands included in the survey were brands that were already known to the respondents of the survey. To adjust for the negative skew, we conducted a log transformation for the two items of brand awareness and used the transformed data for the analysis. Brand awareness and image are combined to construct a second-order variable of brand equity.

### 4.2.3. Financial Performance of Franchisors and Franchisees

To examine the differential effect of franchise operation on franchisors and franchisees, financial data was collected separately. Specifically, we collected data on the annual revenue of each franchisor from the Franchise Disclosure Document and confirmed the information with the annual reports from KIS-VALUE. As for the franchisee performance, we collected the average annual revenue per franchisee provided by the disclosure document. Because the numbers varied over a wide range across industries and firms, we log-transformed the data for analysis.

## 5. Empirical Results

### 5.1. Measurement Model

A confirmatory factor analysis (CFA) was conducted using AMOS 21 to assess the reliability and validity of the two dimensions of brand perception. CFA is a technique to verify how many factors are expected, which factors are related to each other and which items are related to each factor [56]. We can test the hypothesized structure and check if the hypothesized model adequately fits the data through this analysis. All coefficients were found to be statistically significant ($p < 0.05$) and the goodness-of-fit indices were acceptable ($\chi^2 = 5.409$, GFI = 0.992, AGFI = 0.969, SRMR = 0.019, RMSEA = 0.037, CFI = 0.998, and IFI = 0.998). Furthermore, the Average Variance Extracted (AVE) and the Construct Reliability (CR) were calculated and compared with a Squared Multiple Correlation (SMC) to test convergent and discriminant validity. Table 2 provides the computation results of AVE and CR. The AVE and CR exceeded the minimum requirements of 0.5 and 0.7, respectively, showing that there was a convergent validity among the observed variables for each of the brand perception factors. The AVE was also greater than the SMC, indicating that there was a discriminant validity across the two brand equity dimensions [57]. In addition, the values for Cronbach's α were 0.638 and 0.887, which indicated that there was reliability among the indicators.

**Table 2.** Reliability and validity test for brand perception.

| Construct | Indicator | Convergent Validity | |
|---|---|---|---|
| | | AVE | CR |
| Brand Awareness | Recognition Recall | 0.716 | 0.834 |
| Brand Image | Reliable Unique Value | 0.729 | 0.890 |
| Discriminant Validity (AVE > SMC) | | | |
| Construct 1 | Construct 2 | Correlation | SMC |
| Brand Awareness | Brand Image | 0.321 | 0.103 |

CR: Construct Reliability, AVE: Average Variance Extracted, SMC: Squared Multiple Correlations. Convergent Validity: AVE > 0.5, CR > 0.7. Discriminant Validity: AVE > SMC. Both convergent validity and discriminant validity are confirmed.

### 5.2. Structural Model

The structural equation model was estimated using AMOS 21. The goodness-of-fit statistics showed that the model was acceptable ($\chi^2$ (20) = 54.170, GFI = 0.958, AGFI = 0.905, SRMR = 0.045, RMSEA = 0.082, CFI = 0.972, and IFI = 0.973). The results of the hypotheses testing are shown in Figure 2 and Table 3.

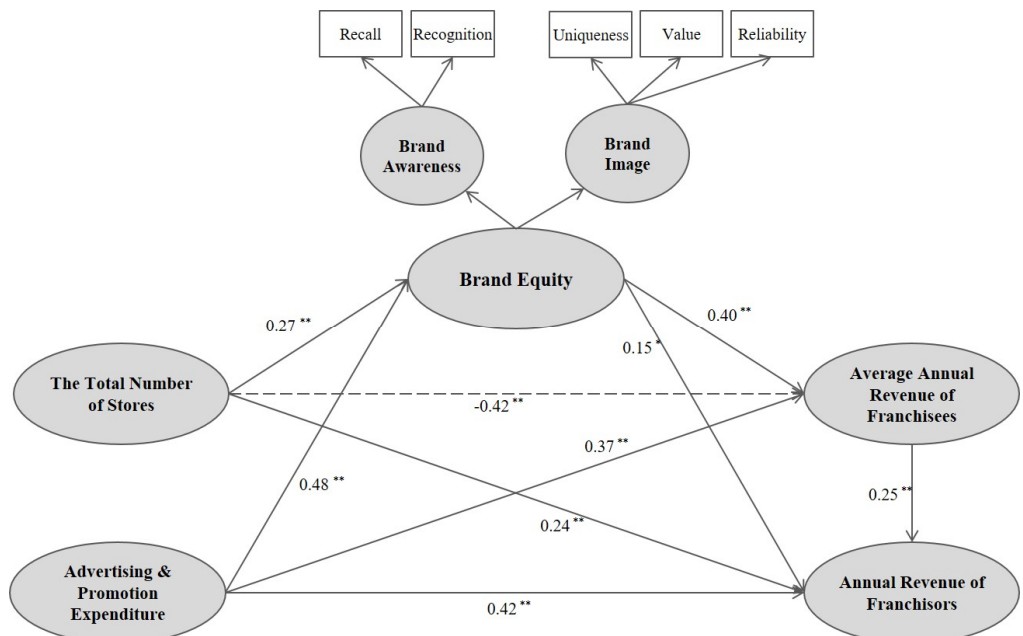

**Figure 2.** Results of the structural equation modeling. * indicates $p < 0.05$; ** indicates $p < 0.01$. The dashed line indicates the path with a path coefficient that is statistically significant but opposite in sign from the hypothesized relationship.

**Table 3.** Hypotheses test results.

| Hypothesized Relationships | | Std. Est. | *t*-Value | Conclusion ($p < 0.05$) |
|---|---|---|---|---|
| Relationships between Franchise Characteristics and Brand Equity | | | | |
| H1a: number of stores → brand equity | + | 0.274 | 2.912 | Supported |
| H1b: advertising and promotion expenditure → brand equity | + | 0.477 | 3.703 | Supported |
| Relationships between Brand Equity and Financial Performances | | | | |
| H2a: brand equity → the average annual revenue of franchisees | + | 0.396 | 3.073 | Supported |
| H2b: brand equity → the annual revenue of the franchisor | + | 0.147 | 1.973 | Supported |
| Relationships between Franchise Characteristics and Financial Performances | | | | |
| H3a: number of stores → the average annual revenue of franchisees | + | −0.416 | −5.707 | Not Supported |
| H3b: number of stores → the annual revenue of the franchisor | + | 0.236 | 4.454 | Supported |
| H4a: advertising and promotion expenditure → the average annual revenue of franchisees | + | 0.369 | 4.522 | Supported |
| H4b: advertising and promotion expenditure → the annual revenue of the franchisor | + | 0.420 | 7.580 | Supported |
| Relationship between Franchisor and Franchisee Performances | | | | |
| H5: the average annual revenue of franchisees → the annual revenue of the franchisor | + | 0.253 | 5.689 | Supported |

For the relationships between franchise marketing decisions and brand equity, the result for H1a shows that the number of total stores has a positive relationship with brand equity. This demonstrates that if there are more stores in operation, then it would be more likely for consumers to become exposed to the franchise brand and form brand images that are more specific and positive, which increases the level of brand equity. Similarly, advertising and promotion expenditures also contribute to building brand awareness and brand images, and ultimately brand equity (H1b). In particular, advertising and promotion

expenditures have a greater influence on brand equity ($\beta = 0.274$) than the number of stores ($\beta = 0.477$). This is partly because advertising and promotion activities involve more direct efforts of franchisors to build or boost brand equity while influencing brand equity via increasing the number of stores in operation has a more subtle impact on consumers' perception of brand awareness and brand image [52].

As for the relationship between brand equity and franchisees' and franchisors' performances, H2a and H2b are both supported, showing that consumers' perception of brand equity has a significant impact on both franchisees' and franchisors' performances. These results imply that consumers are more likely to purchase goods or services of a brand that they can recall and recognize easily and they have constructed specific associations with. Especially, we find that brand equity had a greater effect on franchisees' performance ($\beta = 0.396$) than franchisors' ($\beta = 0.147$). This is because it is usually with the franchisees that consumers make purchases based on their perception of brand equity. In other words, it is highly likely that the purchases that consumers make are directly captured by the franchisees' revenues but indirectly by franchisors' revenue.

Regarding the relationships between franchise marketing decisions and performances of franchisors and franchisees, first, we find that the number of stores positively affects the franchisors' revenue (H3b). This implies that an increase in the number of stores would allow franchisors to collect royalty fees from a greater number of sources, increasing their annual revenues. Unexpectedly, however, the number of stores is found to have a significant negative impact on franchisee performance ($\beta = -0.416$) (H3a). This indicates that as the total number of stores in operation for a franchise brand grows, the average revenue that each franchisee earns decreases. This supports the view that when the number of total stores in operation increases, the geographical density of outlets for a franchise system would increase as well as the internal competition against other franchisees [49]. Subsequently, the overall amount of revenue that each franchisee can earn would decrease. Such negative valence of the effect is opposite to the positive valence of the effects we found for H1a and H2a in which the number of stores has a significant positive effect on brand equity (H1a) and brand equity perception further has a positive influence on franchisees' performance (H2a).

To further investigate the impact of the number of total stores on the franchisee performance, we aggregate the direct effect and the indirect effect via brand equity perception. The bootstrapping analysis (based on 5000 samples) showed that the direct effect of the number of stores on franchisee performance was negatively significant ($\beta = -0.416$, SE = 0.059, 95% CI = [$-0.537$, $-0.307$]). The indirect effect through the brand equity perception was also significant but positive in terms of the coefficient ($\beta = 0.109$, SE = 0.036, 95% CI = [0.050, 0.197]). Most importantly, the total effect of the number of stores on franchisee performance was significantly negative ($\beta = -0.307$, SE = 0.056, 95% CI = [$-0.417$, $-0.201$]). The results indicate that although increasing the number of stores may benefit franchisees through the enhanced brand equity, it incurs greater harm on the franchisee performance by increasing the internal competition with other franchisees.

Next, we find that advertising and promotion expenditures positively influence both franchisees and franchisors' revenues (H4a and H4b). In the case of advertising and promotion expenditure, an increase in the spending on advertising and promotion activities would stimulate consumer purchases and drive performances of both franchisors and franchisees. Lastly, H5, which represents the relationship between the financial performances of franchisees and franchisors, is supported. It indicates that the performances of franchisors and franchisees are significantly related. Specifically, a positive standardized estimate for the path coefficient ($\beta = 0.253$) implies that franchisors' revenues would increase when franchisees earn more revenue. Most of the companies in our dataset have a contractual relationship where the major part of the franchisors' revenue was from the royalty fees, which is determined by the level of annual revenue by franchisees and the placement of orders for ingredients or resources by franchisees in the food and beverage industries. Subsequently, our results show that the franchisees' success transcended to

franchisors, providing evidence that franchisees' and franchisors' performances are closely tied together.

In sum, it can be concluded that the decision regarding the number of stores is a double-edged sword that can harm franchisees' performance while benefiting franchisors' performance. Hence, franchisors should make decisions regarding channel expansion with caution.

## 6. Discussion

### 6.1. Theoretical and Managerial Implications

This research contributes to the prior literature on franchising in mainly two ways. First, we extend the prior literature by examining the impacts of franchise marketing decisions on the financial performances of both franchisors and franchisees. While prior literature on franchising mostly focused on the issues of controlling and managing franchisees from franchisors' perspective [7], there is little research that examines how franchise marketing-decisions influence the performances of franchisors and franchisees. Through this research, we suggest that marketing-decisions can be utilized as a means of achieving balanced profitability that would benefit the franchise relationship. Although the relationships between marketing decisions and firms' performances are well established in the prior literature, it is necessary to examine in the franchise context as well because it is unique in that the interests of franchisors and franchisees are closely tied together. In doing so, we incorporate the viewpoint that the franchise relationship is built for and built based on franchise brand equity (i.e., the cultivation of brand-centric relationship) [9]. Despite its importance, the literature on franchise is yet to delve into the role that brand equity plays in the franchise relationship. Through this research, we add to the existing research by investigating how marketing decisions influence the performances of franchisors and franchisees through brand equity based on the Brand Value Chain model [27].

Second, we add to the previous franchise literature by investigating the nature of franchise relationships in terms of operational performance using actual market data. While prior literature explored various means and approaches for franchise relationship management, they did not show how each of franchisors' and franchisees' performances is influenced as the result of such decisions [58,59]. By analyzing actual performance data that distinguishes performances of franchisors and franchisees, we demonstrate how franchise marketing-decisions drive franchise performances. This research also provides important practical implications regarding the management of franchise relationships. To franchisors in charge of franchise marketing decisions, we suggest that although increasing the number of stores in operation may benefit franchisors in the short run, it also bears the risk of damaging their relationships with franchisees in the long run. It is because franchisees' performances would likely be harmed due to the increased internal competition when there are more stores in operation [49]. If franchisees were not gaining enough profit, they would exit the franchise relationship by not extending the contract [1]. This would negatively influence franchisors in return. Thus, franchisors need to take caution when making decisions about how fast they would expand the franchise system and how many stores they are going to operate.

Conversely, from the franchisees' perspective, we suggest that franchisees should carefully analyze the current status of the franchise operation when they consider opening a franchise chain. As our findings suggest, information about the marketing-decisions would indicate how much support and benefit they can receive from the franchisors. In particular, franchisees should pay special attention to information about the number of stores. Franchisors often attempt to appeal to potential franchisees by emphasizing that their large number of stores is proof of how popular the brand is among other franchisees. However, having a large number of stores in operation would not always be beneficial for franchisees, as it would likely increase the internal competition among franchisees. Thus, potential franchisees must understand that the size of the franchise network may act as a double-edged sword and carefully scrutinize their decisions to start franchise contract.

Most of all, our findings imply that franchisors and franchisees should recognize the importance of protecting and developing the franchise brand together since their franchise relationships are built upon brand equity [9]. Damages to the franchise brand equity would harm both franchisors and franchisees; franchisors would lose potential franchisees and the royalty fees that they would have collected whereas franchisees would lose sales and revenues since consumers would be less likely to purchase their stores [10]. In many cases, franchisees often sabotage the equity of a franchise brand both intentionally and unintentionally by delivering bad quality products and services or not aligning with the franchise brand. However, as our results show, the impact of brand equity on performances is greater for franchisees than franchisors. Because it is the franchisees that consumers directly encounter and make purchases with, consumers' perception of brand equity would be more closely related to franchisees' performance than that of franchisors'. Thus, they need to cooperate with the franchisors and their marketing decisions to enhance brand equity [9].

One part in which such cooperation can be sought after is the advertising and promotional activities because, as our findings show, increasing the spending on advertising and promotional activities has a greater effect on brand equity perception than increasing the number of stores. This indicates that because the decisions regarding advertising and promotional activities are more directly and intentionally geared toward attaining specific brand equity goals than the strategic decisions about the number of stores, the degree of impact is stronger. Thus, franchisors should carefully craft the advertising and promotion strategies for enhancing the franchise brand equity whereas franchisees should aim for communicating messages that are consistent with the franchise brand.

### 6.2. Limitations and Future Research

This research has limitations that need to be addressed through future research. First, the data set in our study is focused on the Korean market. Taking into account the differences in the government regulations and market conditions across countries, the relationships from the estimated model may differ from those in this research [11]. Thus, it is necessary to test the model using data from other countries to enhance the robustness of our findings. For example, French Franchise Federation issues Toute la Franchise (https://www.toute-la-franchise.com, accessed on 22 October 2019) every year, which contains basic information on franchise operations for potential franchisees, such as the age and size of the franchisor, contract type, and the cost of opening a franchise store. Combining it with the financial performance information that is available through databases such as DIANE, we can test the model for other countries.

Second, there may be a selection bias for the franchise brands selected for the analysis in this research. This research uses data samples that appear in all three data set including brand perception data from KMAC, financial data from KIS-VALUE, and the franchise information from KFTC. In particular, the brand perception data from KMAC selectively includes brands with relatively higher shares of each category. Likely, the franchise brands that are smaller in terms of market share are left out from the analysis. Thus, future research needs to accumulate more data for brands varying in size and industry and test them for external validity.

Third, we focus on the short run impact of franchise operations on franchise performance in this study. We matched the data from different sources based on the year it was collected and analyzed the relationships between variables within the same year. However, it is possible that the elements of franchise operations, such as marketing and promotion expenditures, have a more meaningful influence on franchise performance in the long run rather than in the short run by changing the consumers' perception of the franchise brand or the service itself. Thus, future research needs to examine the carryover effect of a franchise operation on the franchise performance in the following years.

Fourth, we do not distinguish between the types of stores in our analysis in this research. However, firms in franchising usually operate franchisee-owned stores and

company-owned stores simultaneously. The two forms of stores differ in terms of the perception of ownership and the degree of commitment to store performance [44]. Thus, in future research, it would be interesting to investigate how franchisors' approach to managing the franchise relationship differs depending on the proportion of franchisee-owned stores within the franchise operation.

Fifth, because the actual market data we utilized for this study is a secondary data, the variables that are included in our research model were limited to the variables that were included in the data we acquired. Subsequently, not all factors that influence brand equity and franchise performance is taken into consideration in this study. One of those variables is the perceived level of service quality [60]. Hence, in future research, it would be necessary to investigate the role that perceived service quality plays on franchise brand equity and performance. In particular, because perceived service quality is a multidimensional construct as suggested by the five dimensions of SERVQUAL [60], consumer perceptions of franchise service quality can be measured on these dimensions and analyzed to explore which of the dimensions has a greater effect on franchise brand equity and performance.

Lastly, this study performed CFA to examines the effect of marketing decisions on brand equity and franchise performance, but did not try analysis methods using online platforms such as Twitter [61]. In this study, we analyzed using actual market and customer data, but using the analysis of user-generated content (UGC) on online platform will allow us to look at the perception of the brand equity from a more macroscopic perspective and the impact on franchise performance. In future studies, we believe that looking at the opinions of franchisors and franchisees on brand equity using big data extracted from online platforms will be an interesting field of sustainable franchise study.

**Author Contributions:** All authors worked collectively and significantly contributed to this work. Conceptualization, E.L., J.-H.K. and C.S.R.; methodology, E.L. and J.-H.K.; software, E.L.; validation, J.-H.K. and C.S.R.; formal analysis, J.-H.K. and C.S.R.; investigation, E.L.; data curation, J.-H.K.; writing—original draft preparation, C.S.R. and J.-H.K.; writing—review and editing, E.L. and C.S.R.; supervision, C.S.R.; project administration, E.L. All authors have read and agreed to the published version of the manuscript.

**Funding:** This research received no external funding.

**Institutional Review Board Statement:** Not applicable.

**Informed Consent Statement:** Not applicable.

**Data Availability Statement:** In this study, we collected the franchise data from Franchise Disclosure Document (FDD), and financial performance data from KIS-Value respectively. FDD announced by the Korean government is a report that franchisors are required to disclose with the goal of protecting the rights of potential franchisees and reduce the effect of information asymmetry. The report must include detailed information about the chain's current franchise operation from basic information. KIS-VALUE is a financial information database provided by NICE Information Service Co., Ltd., a subsidiary of the Korean credit evaluation company NICE. It provides various types of corporate information that includes annual reports, financial information, credit records, and audit reports of Korean companies.

**Acknowledgments:** We acknowledge Ki-Dong Lee, the director of Business Value Department in KMAC for providing brand equity data (K-BPI).

**Conflicts of Interest:** The authors declare no conflict of interest.

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
