# Peer review of "Effects of Marketing Decisions on Brand Equity and Franchise Performance"

_sustainability, doi:10.3390/su13063391_

Round 1

Reviewer 1 Report

Interesting job enjoyed to look at

Reviewer 2 Report

I would like to start by thanking the opportunity given to me by being able to review this research paper. I hope the authors find my recommendations productive and that it help them improve their research work.
The objective of the study is clear, the authors intend to provide a mode of action franchising companies to achieve sustainable profitability between franchisors and franchisees, through decisions related to the value of the branding, financial returns, and making efficient marketing choices.
The document is based on up-to-date scientific research. I also know that it implements the discipline of marketing to a business sector that still needs to be studied and analyzed in terms of marketing terms and sustainability relationship between
franchisors and franchisees.
The five hypotheses proposed by the authors seem correct to me and allusion is made to them throughout the article filling in the research gaps from past research.
The confirmatory factor analysis used as a model to measure the hypotheses seems correct since the objective of the article is the "confirmation" of the theory. I recommend the reading and citation of the following article “Lloret-Segura, S., Ferreres-Traver, A., Hernández-Baeza, A., & amp; Tomás-Marco, I. (2014). The exploratory factor analysis of the items: a guide practical, revised and updated. Anales de Psicología / Annals of Psychology, 30 (3), 1151-1169 ”to reform the theoretical framework on the decision of the selected factor analysis.
However, about the graph used to represent the structural relationships. Besides, I would have exposed more models to support the premises visually.
The use of the AMOS program seems to me adequate for the hypotheses when measuring how the behavior of some variables affects others. In this sense i suggest the authors to review and cite Reyes-Menendez, A., Saura, J. R., & Martinez-Navalon, J. G. (2019). The impact of e-WOM on hotels management reputation: exploring tripadvisor review credibility with the ELM model. IEEE Access7, 68868-68877.
Highlight the correct analysis carried out for the reliability and validity of the dimensions of the brand perception, as well as the metrics used to calculate the adjustment.
The first hypothesis referring to “The number of total stores has a positive effect on the brand value "and" Spending on advertising and promotion has a positive effect on the value of the brand ”, is well defined and explained both in the theoretical framework and in the results.
Regarding the second premise that alludes to the positive effects of the value of the brand in franchisors and franchisees, I agree with the variables chosen for the brand awareness measurement: recall and awareness. Although when treating the franchise research, perhaps I would have chosen yet another variable such as quality perceived by the consumer to analyze if all franchises have the same quality perceived or not, I think it would be an interesting aspect to analyze.
Also in this second hypothesis to assess the knowledge and image of the brand, I would propose the authors to consider performing a User Generated Content Analysis (UGC), to find out what they think of the brand and analyze it. For this type of analysis or to deny its use, I propose the following reading “Reyes-Menendez, A., Saura, J. R., & amp; Filipe, F. (2020). Marketing challenges in the # MeToo era: Gaining business insights using an exploratory sentiment analysis. Heliyon, 6 (3), e03626 ".
I think the authors have done a good job on the third hypothesis, especially on the H3a when trying to address and verify negative results by proposing other methods like the one presented in the research (direct and indirect effect). The conclusion that state in this premise is clear "The results indicate that although increasing the number of stores may benefit franchisees through the enhanced brand equity, it incurs greater harm on the franchisee performance by increasing the internal competition with other
franchisees”.
Regarding the fourth and fifth hypotheses referring to spending on advertising and promotion and the positive effect of the financial results of the franchisees on the franchisors, are clearly and correctly justified.
Regarding the structure, an important aspect to be taken into account by the authors, would be the absence of conclusions that made reference to the chosen methods, as well as the main results collected.
Good conduct of the discussion supported by both your research and that of others authors.
The four limitations raised as well as their possible future research seem very interesting to analyze.
The first one referred to the fact that the data collected in this research only belong to Korean franchises, it seems very important to me since, as the authors indicate, results may vary significantly using data from other countries.
Regarding the second limitation, effectively the results can be altered depending on whether it is a franchise

Reviewer 3 Report

The topic of the article is not new, but well developed and has an applied nature.  The  aim of revealing the nature of the marketing decision in a franchise relationship involving two closely related parties is duly provided for. The model developed by the authors, which examines the relationship between marketing decisions on channel intensity and advertising and advertising costs, brand ownership, and the financial performance of franchisees and franchisors, is clear and well-focused.

Empirical research and its substantiation are positively evaluated:

  1. Examine the impact of franchise marketing decisions on the financial performance of both franchisors and franchisees;
  2. Analyzes a separate set of information as financial indicators of franchisors and franchisees. This allowed for a sufficiently broad study of how franchisors and franchisees are affected differently by channel intensity or advertising and promotional activities Demonstrate how franchise marketing decisions drive franchise results;
  3. Suggests evaluating consumers who are directly exposed to franchise services because their understanding of brand ownership is more related to the franchisee. The authors ’decisions on closer collaboration in the franchise business for both sides to strengthen brand ownership are logical.
  4. The suggestion that marketing solutions can be used as a means to achieve balanced profitability that would benefit the franchise relationship is acceptable;

The work (article) is well documented.

Discussion question on hypothesis H1b: The advertising and promotion expenditure has a positive effect on brand equity. It has long been proven the importance of its presentation to be considered. Although, as the authors write, it was necessary to make sure that there was a link between advertising and advertising costs and brand ownership in the franchise business. However, in the service businesses in question, this has been demonstrated by other authors, including those cited in the literature. Perhaps the authors could supplement the importance of presenting this hypothesis in this article.

Round 2

Reviewer 2 Report

Dear authors, i have reviewed your paper and i can confirm that my comments were properly addressed